# The role of problem-solving skills in the prevention of suicidal behaviors: A systematic review and meta-analysis

**Nahid Darvishi[1,2], Mehran Farhadi[3], Bita Azmi-Naei[4], Jalal Poorolajal[4,5]***

**1** Department of Psychology, School of Human Sciences, Sanandaj Branch, Islamic Azad University, Sanandaj, Iran, **2** Consultation Center, Department of Education, Hamadan, Iran, **3** Department of Psychology, Faculty of Economics and Social Sciences, Bu-Ali Sina University Hamadan, Hamadan, Iran, **4** Department of Epidemiology, School of Public Health, Hamadan University of Medical Sciences, Hamadan, Iran, **5** Modeling of Noncommunicable Diseases Research Center, Hamadan University of Medical Sciences, Hamadan, Iran

* poorolajal@umsha.ac.ir

## Abstract

### Background

This meta-analysis was conducted to assess the association between problem-solving skills and suicidal behaviors and elucidate the potential role of problem-solving skills in influencing the occurrence of suicidal behaviors.

### Methods

PubMed, Web of Science, and Scopus were searched until August 16, 2023. Studies addressing the associations between problem-solving skills and suicidal behaviors were included. The $I^2$ statistics were used to examine between-study heterogeneity. The Begg and Egger tests were used to determine the possibility of publication bias. Using a random-effects model, the overall effect size was presented as an odds ratio (OR) or standard mean difference (SMD) with 95% confidence intervals (CIs).

### Results

Of 8040 identified studies, 29 (including 974,542 participants) were eligible. Based on observational studies, problem-solving skills were found to be inversely related to suicidal ideation (OR = 0.64; 95% CI: 0.50, 0.82); suicide attempts (OR = 0.75; 95% CI: 0.63, 0.89), and suicide death (OR = 0.02; 95% CI: 0.01, 0.03). The overall score of problem-solving skills was higher in those who did not attempt suicide than those who did (SMD = 0.84; 95% CI: 54, 1.13). Based on randomized clinical trials, problem-solving therapy was found to reduce the risk of suicide (OR = 0.51; 95% CI: 0.29, 0.87). Furthermore, the overall risk of suicide was lower among those who received problem-solving therapy than those who did not (SMD = -0.02; 95% CI: -0.29, 0.25).

**Funding:** The authors received no specific funding for this work.

**Competing interests:** The authors have declared that no competing interests exist.

## Conclusions

This meta-analysis revealed an inverse association between problem-solving skills and suicidal behaviors. However, further research is needed to better understand the complex relationship between problem-solving skills and suicidal behaviors.

## Introduction

Suicide is a significant problem for public health worldwide. Every year, more than 700,000 individuals die by suicide throughout the world [1]. Additionally, there are around 25 suicide attempts for every suicide [2]. Suicide is one of the main causes of death worldwide, accounting for more deaths than homicide, war, breast cancer, malaria, and HIV/AIDS [1].

Every 40 seconds a person dies due to suicide somewhere in the world [1]. However, suicides are preventable. One of the strategies suggested for the prevention of suicide is implementing universal and targeted school-based socio-emotional learning programs to help adolescents with problem-solving and coping abilities [3]. Problem-solving is a process that involves identifying and defining a problem, determining its underlying causes, generating and prioritizing solutions, implementing a chosen solution, and evaluating the results [4]. This process can be broken down into five key steps, including: (a) defining the problem, (b) identifying the causes of the problem, (c) generating potential solutions, (d) implementing the chosen solution, and (e) evaluating the effectiveness of the solution. This step-by-step approach to problem-solving can be used in a variety of settings, from personal decision-making to complex organizational problem-solving.

Suicidal behaviors are a major public health concern, with devastating consequences for individuals, families, and communities [5]. Despite extensive research on the topic, there is still much to learn about the factors that contribute to suicidal behaviors and how they may be prevented. One potential protective factor for suicidal behaviors is problem-solving ability, which involves the ability to identify and implement effective solutions to problems. However, the relationship between problem-solving ability and suicidal behaviors is complex and multi-faceted, with different types of problem-solving abilities potentially relating to different types of suicidal behaviors. For example, deficits in emotion regulation or cognitive flexibility may be associated with suicidal ideation, while deficits in problem-solving ability may be more strongly related to suicide attempts [6–8]. Understanding these different types of problem-solving abilities and how they relate to suicidal behaviors is essential for developing effective prevention and intervention strategies.

Past studies have investigated the relationship between problem-solving ability and suicidal behaviors, but the results have been mixed and conflicting. Some studies have found that high problem-solving ability is associated with a lower risk of suicidal behaviors [7, 9–13], while others have found no significant association or even a positive association between problem-solving ability and suicidal behaviors [14–16]. Some interventional studies have shown that improving problem-solving skills can reduce the risk of suicide [17, 18]. These conflicting results may be due to differences in study design, population characteristics, and measurement tools.

Given the variation in existing work in terms of definitions, measurement, and populations, it is challenging to directly compare existing studies and draw clear conclusions about the relationship between problem-solving ability and suicidal behaviors. Therefore, this study aims to conduct a meta-analysis of existing studies to provide a comprehensive understanding of the

association between problem-solving ability and suicidal behaviors. The broad focus of this study is an initial step towards finding points of consensus and difference between existing studies while acknowledging the challenges of comparing studies with different methodologies and definitions. By synthesizing the existing evidence, this study aims to advance our understanding of the relationship between problem-solving ability and suicidal behaviors and to inform the development of effective prevention and intervention strategies.

## Methods

### Eligibility criteria (PICOS)

**Population.** The population of interest for this study was broad and included diverse groups such as the general population, students, veterans, inmates, workers, and patients with mental illness.

**Intervention/Exposure.** The exposure of interest was problem-solving skills regardless of the tools used to measure the skills. Problem-solving skills are the ability to identify, analyze, and solve problems in a logical and systematic way. Effective problem-solving involves several steps, including defining the problem, gathering information, analyzing the problem, developing solutions, implementing the chosen solution, and evaluating the results [19]. The assessment of "problem-solving skills" was carried out through the utilization questionnaires that have been developed and validated for the purpose of evaluating problem-solving abilities. A comprehensive list of these questionnaires is provided in Table 1.

**Control.** In cases where studies categorized individuals based on their problem-solving skills (high vs. low), we examined the odds of suicidal behaviors in those with high problem-solving skills compared to those with low problem-solving skills (reference group). This allowed us to assess the impact of varying problem-solving skill levels on suicidal behaviors. In cases where studies measured the mean problem-solving skill scores among individuals with and without suicidal behaviors, we calculated the standardized mean difference (SMD) of the problem-solving skill scores between these two groups. This approach enabled us to explore the magnitude of difference in problem-solving abilities between individuals exhibiting suicidal behaviors and those who did not.

**Outcome.** The primary outcome of interest in this study was various types of suicidal behaviors, including suicidal ideation, suicide plans, suicide attempts, and suicide death. Suicidal ideation, suicide plans, and suicide attempts were assessed by taking a history from study participants. Information on suicide death was obtained from medical records or death certificates.

**Studies.** Regardless of publication status or language, we included observational studies (cross-sectional studies, case-control studies, and cohort studies) as well as randomized clinical trials (RCTs) addressing the association between problem-solving skills and suicide behaviors (suicidal ideation, suicide attempts, and suicide death). Studies that provided the data required to report the effect size in the form of odds ratio (OR), hazard ratio (HR), or mean difference (MD). The results of these different studies were pooled and reported separately.

### Search methods

PubMed, Web of Science, and Scopus were searched until August 16, 2023. The reference lists of all retrieved studies and pertinent reviews were also screened for additional references. The following keywords were searched: (problem solving or coping skills or resilience skills or solution-oriented skills or cognitive flexibility or decision-making abilities or critical thinking or creative thinking or analytical thinking) and (suicide or suicidal or suicidality) [S1 Table].

**Table 1. Characteristics of the included studies.**

| 1st author, yr | Country | Study population | Age mean/ range | sex | Study design | Tools | Suicide type | Suicide time | Sample size | Score of quality |
|---|---|---|---|---|---|---|---|---|---|---|
| Akbari 2015 | Iran | General | 25.51 | Both | Case-control | BMCQ | Attempt | Past month | 600 | ******* |
| Angora 2022 | Spain | General | 35–65 | Both | RCT | Unspecified | Attempt | Past year | 649 | **** |
| Biggam 1999 | UK | Inmates | 18.80 | Male | Case-control | MEPS | Attempt | Past month | 40 | ***** |
| Burke 2016 | USA | General | 12.39 | Both | Cohort | CRSQ | Ideation | Past year | 324 | ******* |
| Dieserud 2002 | Norway | General | 18+ | Both | Case-control | PSI | Ideation | Past month | 321 | ******* |
| Donald 2006 | Australia | General | 18–24 | Both | Case-control | PPSI | Attempt | Past month | 475 | ******* |
| Eidhin 2002 | UK | Inmates | 25.29 | Male | Case-control | MEPS | Attempt | Past month | 31 | ***** |
| Eskin 2008 | Turkey | General | 19.10 | Both | RCT | SPS | Ideation | Past month | 121 | ****** |
| Fitzpatrick 2005 | USA | General | 19.02 | Both | RCT | BSS | Ideation | Past month | 110 | **** |
| Foroughipour 2013 | Iran | Epilepsy | No data | Both | Cross-sectional | QCM | Ideation | Past month | 74 | ****** |
| Fried 2013 | USA | General | 14.86 | Both | Cohort | Unspecified | Attempt | Past month | 3376 | ****** |
| Gururaj 2004 | India | General | 15–69+ | Both | Case-control | Unspecified | Death | Lifetime | 538 | ****** |
| Howat 2002 | UK | General | 74.75 | Both | Case-control | MEPS | Attempt | Past month | 40 | ****** |
| Kaviani 2004 | Iran | General | 27.80 | Both | Case-control | MEPS | Attempt | Past month | 40 | ****** |
| Kidd 2007 | Canada | General | 14–24 | Both | Cross-sectional | WCQ | Attempt | Lifetime | 208 | ******** |
| Lannoy 2022 | Sweden | General | 42.60 | Both | Cohort | SMES | Attempt | Past month | 951,618 | ******** |
| Li 2012 | China | General | 26.23 | Both | Case-control | CRI | Attempt | Past month | 808 | ******** |
| Linda 2012 | USA | General | 19.00 | Both | Case-control | MEPS | Attempt | Lifetime | 96 | ****** |
| Mostafavi Rad 2012 | Iran | General | 45.41 | Female | Case-control | MFAD | Attempt | Past month | 53 | ****** |
| Nezu 2017 | USA | Veteran | 38.03 | Both | Case-control | SPSI | Attempt | Lifetime | 46 | ******* |
| Pollock 2001 | UK | General | 21–72 | Both | Case-control | MEPS | Attempt | Past month | 48 | ****** |
| Roskar 2007 | Slovenia | General | 43.94 | Both | Case-control | TOL | Attempt | Past month | 50 | ****** |
| Sadowski 1993 | USA | General | 15.02 | Both | Case-control | SPSI | Attempt | Past month | 60 | ****** |
| Sarkisian 2021 | USA | General | 7.70 | Both | Case-control | Lab-TAB | Ideation | Past year | 116 | ******* |
| Shelef 2014 | Israel | Veteran | 19.70 | Both | Case-control | PSI | Attempt | Lifetime | 65 | ****** |
| Sugawara 2012 | Japan | Workers | 40–60 | Both | Cross-sectional | CES-D | Ideation | Lifetime | 6,762 | ******** |
| Tang 2015 | China | Students | 20.85 | Both | Cross-sectional | CRI | Ideation | Past year | 5,972 | ******** |
| Unützer 2006 | USA | Mental | 71.20 | Both | RCT | HSCL | Ideation | Past month | 1,801 | ******* |
| Xavier 2019 | Brazil | General | 17.20 | Both | RCT | ISO | Ideation | Past month | 100 | ******* |

The measurement tools used for measuring problem-solving skills and suicidal behavior are listed in alphabetical order. ACS: Adolescent Coping Scale, BMCQ: Billings and Moos Coping Questionnaire, BSS: Beck Suicide Scale, CES-D: Center for Epidemiologic Studies Depression Scale, CRI: Coping Response Inventory, CRSQ: Children's Response Styles Questionnaire, HSCL: Hopkins Symptoms Checklist, ISO: Inventory of Suicide Orientation, Lab-TAB: Laboratory Temperament Assessment Battery, MEPS: Means-End Problem-Solving, MFAD: McMaster Family Assessment Device, PPSI: Personal Problem-Solving Inventory, PSI: Problem-Solving Inventory, SMES: Swedish Military Evaluation Scale, SPS: Suicide Probability Scale, SPSI: Social Problem-Solving Inventory, TOL: Tower of London, QCM: Questionnaire of Coping Mechanisms, WCQ: Ways of Coping Questionnaire

## Study selection

The search results obtained from electronic databases were merged and duplicates were eliminated using the EndNote program. Two authors evaluated independently the titles and abstracts to determine which studies matched the eligibility criteria for this review. The authors discussed and agreed upon any differences. For more information, we downloaded the complete texts of research that seemed pertinent.

## Data extraction

The extracted data from the relevant studies were entered into an electronic data sheet prepared in Stata software. The following data were extracted: first author's name, year of publication, country, language, age (mean, range), gender, study population (general population, students, veterans, inmates, mental illness), study design (cross-sectional, case-control, cohort, RCTs), measurement tools, suicidal behaviors (suicidal ideation, suicide attempts, suicide death), suicide time (past month, past year, lifetime), sample size, analysis of potential confounders (adjusted, unadjusted), and effect size (OR, HR, MD) with associated 95% confidence interval (CI).

## Methodological quality

The quality of the observational studies was explored using the Newcastle-Ottawa Scale (NOS) [20] and that of RCTs was assessed by the Delphi List [21]. Each study received a maximum of nine stars using either of these tools. Studies were categorized as either high-quality or low-quality based on their star rating, with high-quality studies receiving 7 stars or more [S2 Table].

## Data synthesis

The overall effect size was reported as an odds ratio (OR) or standard mean difference (SMD) with 95% CIs. SMD values of 0.2–0.5 were regarded as small, 0.5–0.8 as medium, and values greater than 0.8 as large [22]. Data were analyzed using Stata software version 14 (StataCorp, College Station, TX, USA) and Review Manager 5. The meta-analysis was performed to create a summary measure using the random effects model at a 0.05 significance level [23].

The random-effects model assumes that the true effect size varies across studies due to differences in study design, population, and other factors. This model takes into account both within-study and between-study variability to produce a more conservative estimate of the effect size and a wider confidence interval.

## Heterogeneity and publication bias

The chi-square ($\chi^2$) test was used to examine study heterogeneity and the tau-square ($\tau^2$) test to estimate the variance between studies [23]. The $I^2$ value [24] was used to assess the possibility of heterogeneity across studies. Based on the $I^2$ value, heterogeneity was classified as low (50%), moderate (50–74%), or high (75%) [24]. We performed a meta-regression analysis to find the potential sources of heterogeneity.

The likelihood of publication bias was evaluated using the Egger [25] and Begg [26] tests and the Trim-and-Fill method.

## Results

### Description of studies

In total, 8040 studies were discovered, including 6713 studies that were found by searching the electronic databases until August 16, 2023, and 1327 articles were discovered by screening the reference lists of the included studies. After excluding duplicates and ineligible studies, 29 studies [7, 9–13, 15–18, 27–45] including 974,542 individuals were included in the meta-analysis (Fig 1). Table 1 provides information about the included studies in more detail.

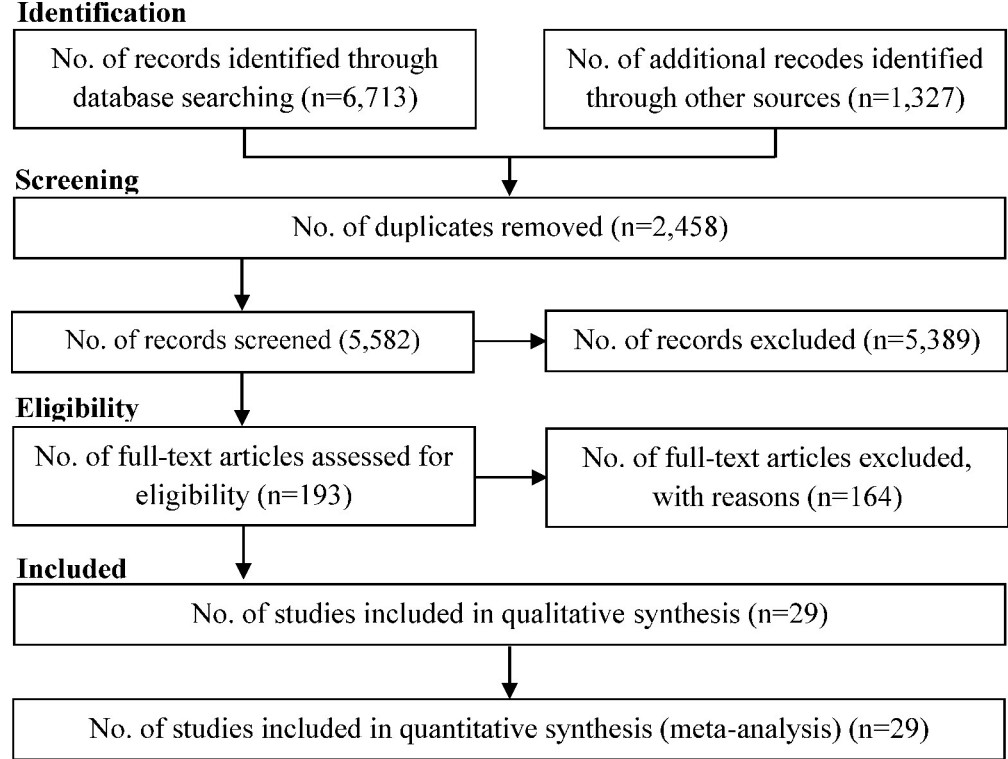

**Identification**

No. of records identified through database searching (n=6,713)

No. of additional recodes identified through other sources (n=1,327)

**Screening**

No. of duplicates removed (n=2,458)

No. of records screened (5,582)

No. of records excluded (n=5,389)

**Eligibility**

No. of full-text articles assessed for eligibility (n=193)

No. of full-text articles excluded, with reasons (n=164)

**Included**

No. of studies included in qualitative synthesis (n=29)

No. of studies included in quantitative synthesis (meta-analysis) (n=29)

**Fig 1. The flow of information through the different phases of the systematic review.**

## Synthesis of results

Based on five observational studies (Fig 2), the overall OR for high versus low problem-solving skills and suicidal ideation was 0.64 (95% CI: 0.50, 0.82). According to the overall effect measure, high problem-solving skills lower the probability of suicide ideation by about 36% ($P$ = 0.0004). Between-study heterogeneity was moderate ($I^2$ = 66%). The Begg test (P = 0.348) and the Egger test (P = 0.636) revealed no evidence of publication bias.

Based on five observational studies (Fig 2), the overall OR for high versus low problem-solving skills and suicide attempts was 0.75 (95% CI: 0.63, 0.89). According to the overall effect measure, high problem-solving skills lower the probability of attempting suicide by about 25% ($P$ = 0.001). Between-study heterogeneity was high ($I^2$ = 99%). The Begg test (P = 0.348) and the Egger test (P = 0.361) revealed no evidence of publication bias.

Based on one observational study (Fig 2), the OR for high versus low problem-solving skills and suicide death was 0.02 (95% CI: 0.01, 0.03). According to this finding, high problem-solving skills lower the probability of suicide death by about 98% ($P$<0.001).

Based on the results of two RCTs (Fig 3), the OR for high versus low problem-solving skills and suicide death was 0.51 (95% CI: 0.29, 0.87). According to this finding, high problem-solving skills lower the probability of suicide death by about 49% ($P$ = 0.010). Between-study heterogeneity was moderate ($I^2$ = 70%).

According to the results of 13 studies (Fig 4), the total score of problem-solving skills was higher in those who did not attempt suicide than those who did (SMD = 0.84; 95% CI: 54, 1.13; P<0.001). That means the overall standard mean score of problem-solving skills in people who did not attempt suicide was 0.84 larger than those who attempted suicide. Between-study heterogeneity was moderate ($I^2$ = 79%).

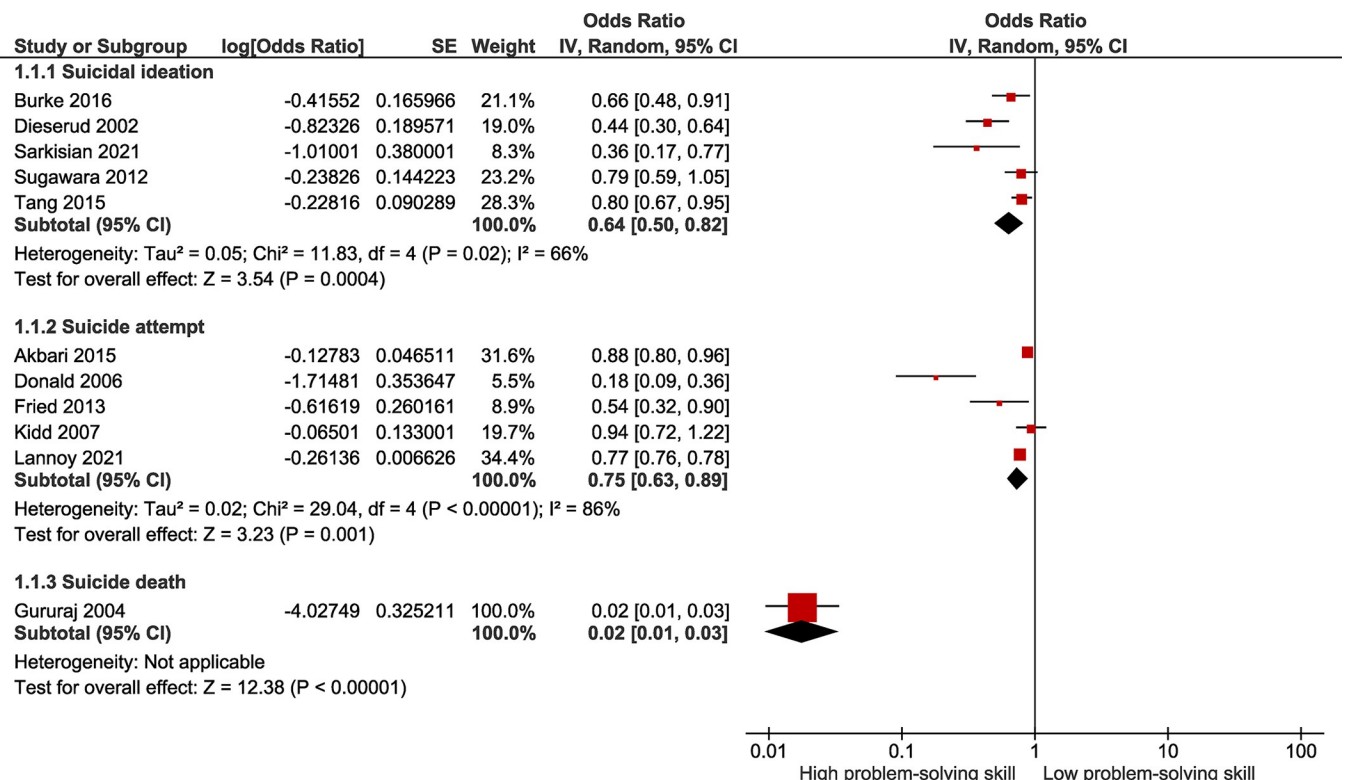

**Fig 2. Forest plot of the association between problem-solving skills and suicidal ideation, suicide attempt, and suicide death.**

According to the results of three RCTs (Fig 5), the total score of risk of suicide was lower in those who received problem-solving therapy versus those who did not (SMD = -0.02; 95 CI: -0.29, 0.25) but the results were not statistically significant (P = 0.870). Between-study heterogeneity was low ($I^2 = 14\%$).

## Meta-regression

We conducted a multivariate meta-regression analysis taking into account several factors including study population, study design, suicide time, and measurement tools (Table 2) to investigate the possible sources of heterogeneity. The results revealed that none of the factors affected the observed heterogeneity.

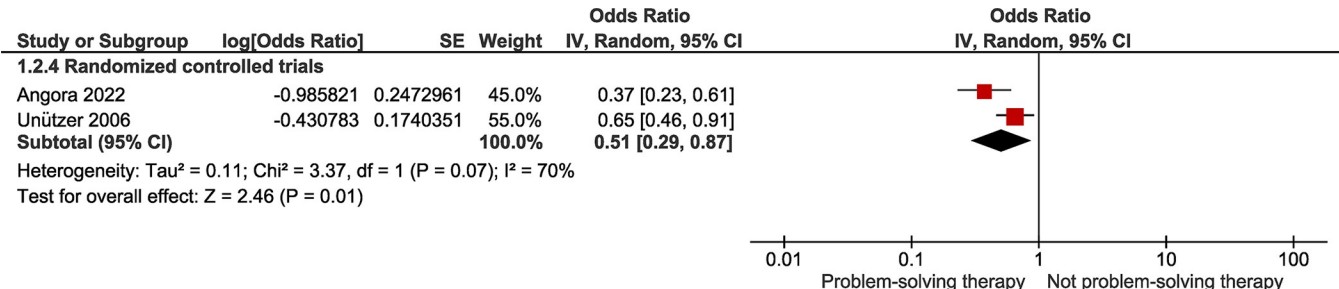

**Fig 3. Forest plot of the association between problem-solving therapy and the risk of suicide.**

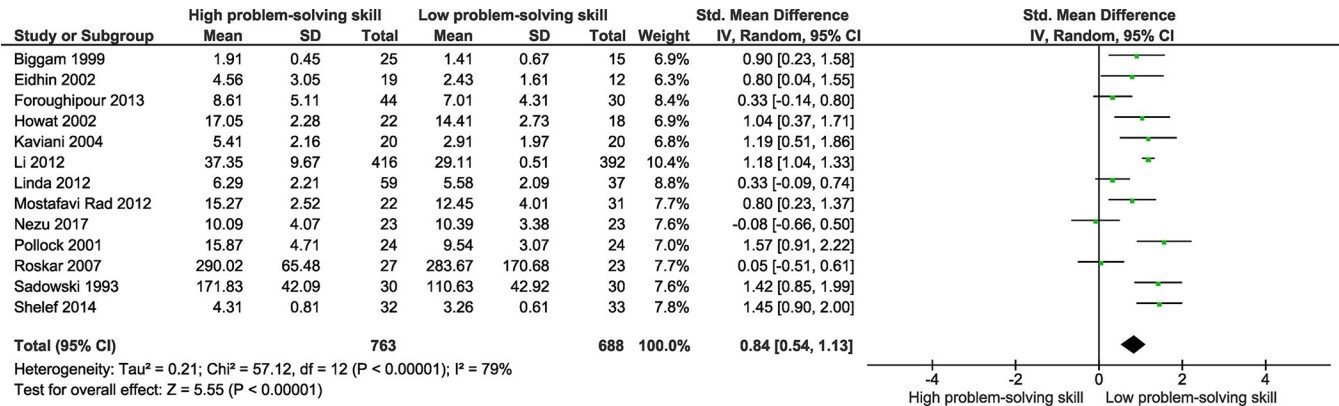

**Fig 4. Forest plot of standard mean difference (SMD) of the total score of problem-solving skills among those who attempted suicide versus those who did not attempt suicide.**

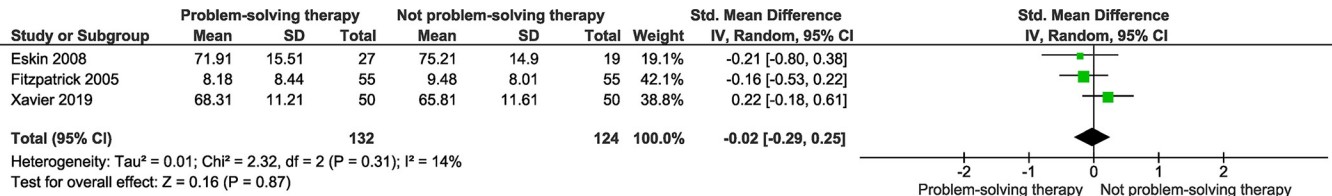

**Fig 5. Forest plot of standard mean difference (SMD) of the total score of risk of suicide among those who received problem-solving therapy versus those who did not receive problem-solving therapy.**

**Table 2. Assessing the source of heterogeneity across studies reporting the association between problem-solving skills and suicidal ideation and suicide attempt using the meta-regression.**

| Suicidal behaviors/covariates | Coefficient | SE | t | P-value | 95% CI | |
|---|---|---|---|---|---|---|
| **Suicidal ideation** | | | | | | |
| Study population | 1.037649 | 0.029803 | 1.29 | 0.421 | 0.720373 | 1.494664 |
| Study design | 0.810272 | 0.094484 | -1.80 | 0.322 | 0.184145 | 3.565335 |
| Suicide time | 1.190113 | 0.228958 | 0.90 | 0.532 | 0.103269 | 13.71531 |
| Measurement tools | 0.995766 | 0.001889 | -2.24 | 0.268 | 0.972050 | 1.020061 |
| Constant | 0.982424 | 0.359994 | -0.05 | 0.969 | 0.009337 | 103.3646 |
| **Suicide attempt** | | | | | | |
| Study population | 3.677053 | 5.454414 | 0.88 | 0.473 | 0.006218 | 2174.425 |
| Study design | 0.661504 | 0.528518 | -0.52 | 0.657 | 0.021261 | 20.58216 |
| Suicide time | 0.377527 | 0.565420 | -0.65 | 0.582 | 0.000600 | 237.4479 |
| Measurement tools | 0.991331 | 0.009826 | -0.88 | 0.472 | 0.949943 | 1.034522 |
| Constant | 2.548539 | 7.327621 | 0.33 | 0.776 | 0.000011 | 601158.0 |

## Discussion

The meta-analysis found that high problem-solving skills were associated with a lower probability of suicidal ideation, suicide attempts, and suicide death and that the total score of problem-solving skills was higher in individuals who did not attempt suicide than those who did. The meta-analysis also found that the total score of problem-solving skills was higher in individuals who did not attempt suicide than those who did and that receiving problem-solving

therapy was associated with a lower risk of suicide, although the latter result was not statistically significant. However, there was high between-study heterogeneity for some of these results, which may limit the generalizability of the findings. In addition, there was evidence of publication bias for some of the results, which suggests that some studies with negative or non-significant findings may not have been published. This may have influenced the overall effect size and could limit the validity of the conclusions.

The results of the meta-regression analysis did not reveal any significant factors that contributed to the observed heterogeneity. This suggests that the variability in the results may be due to differences in study characteristics or other unmeasured factors.

The meta-analysis included 29 studies that varied in terms of study design, population characteristics, and measurement tools. For example, some studies included in the meta-analysis used self-report measures to assess problem-solving skills and suicidal behaviors, while others used clinical interviews or medical records. Additionally, some studies included individuals with specific clinical diagnoses (e.g., students, veterans, inmates, workers, and patients with mental illness), while others included community-based samples. These differences in study design and population characteristics may contribute to the observed between-study heterogeneity in the meta-analysis. For example, some studies may have included individuals with more severe mental health problems or a higher risk of suicidal behaviors, which could influence the observed associations between problem-solving skills and suicidal behaviors. Additionally, differences in measurement tools may have introduced variability in the results, as different tools may assess different aspects of problem-solving skills or suicidal behaviors. Despite these differences, the association between high problem-solving skills and a lower probability of suicidal behaviors suggests that improving problem-solving skills may be an effective strategy for reducing the risk of suicide. However, it is important to consider the potential impact of these differences on the observed associations and to interpret the results in light of the potential biases and limitations of the study.

According to the results of this meta-analysis, problem-solving skills have a protective effect against suicidal ideation, suicide attempts, and suicide death. We must keep in mind, though, that there is not just one single risk or protective factor for suicide. It typically happens as a result of a complex interplay between several risk factors and protective factors [46–49]. As a result, risk and protective factors cannot be broken down into discrete components and must instead be seen as a whole. Risk factors encourage suicide while protective factors deter it. In this regard, if risk and protective variables are equal or if protective factors outweigh risk factors, suicide will not occur [50]. Therefore, the role of problem-solving skills in the prevention of suicidal behaviors should be taken into account along with other influential factors.

One of the important cognitive risk factors connected with suicidal behaviors is problem-solving deficits or difficulties in identifying problems and producing suitable solutions [51, 52]. Lack of problem-solving skills can be characterized by a negative problem orientation (e.g., seeing problems as threats, doubting problem-solving abilities, or a tendency to become frustrated when facing problems); an impulsive/careless style (e.g., attempting to address problems quickly or insufficiently), and/or an avoidant style (e.g., passivity, inaction, and overdependence on others) [53]. Indeed, suicide attempters may be more passive in their problem-solving approach than non-suicidal attempters consistent with an avoidant problem-solving style [54]. On the other hand, some suicide attempters may be characterized by an increased rate of impulsivity and carelessness and a tendency to develop a pessimistic attitude toward problems [55].

Consistent with this meta-analysis, Townsend et al. assessed the effect of problem-solving therapy on deliberate self-harm. They concluded that problem-solving therapy appeared to produce better results than the control group concerning improvement in depression,

hopelessness, and problems [56]. Another systematic review was conducted by Gray et al. to address the applications of problem-solving therapy to prevent suicide in patients with bipolar disorder. They concluded that training problem-solving strategies can be considered either as a part of psychological treatment in patients with bipolar disorder or as a specific preventive intervention [57].

## Limitations

This systematic review was associated with several limitations and potential biases. First, there was high between-study heterogeneity in some of the results, which may be due to differences in study characteristics or other unmeasured factors. This variability may limit the generalizability of the findings. Second, our findings revealed evidence of publication bias for some of the results, which suggests that some studies with negative or non-significant findings may not have been published. This may have influenced the overall effect size and could limit the validity of the conclusions. Third, this meta-analysis included a relatively small number of studies, which may limit the statistical power and generalizability of the findings. Fourth, the studies included in the meta-analysis used different tools to measure problem-solving skills, which may have introduced variability in the results. Fifth, most of the studies included in the meta-analysis were observational studies, which may limit the ability to establish causality. It is important to consider these limitations when interpreting the results and drawing conclusions. Further research is needed to confirm these findings and to address these limitations.

The meta-analysis findings suggest that bolstering problem-solving skills is linked to decreased likelihood of suicidal ideation, attempts, and deaths, prompting health policies to prioritize the advancement of problem-solving skills training. This approach can be integrated into school curricula, workplace wellness initiatives, and community mental health programs. Additionally, the inclusion of problem-solving therapy within mental health treatments, particularly for individuals vulnerable to suicide, warrants consideration. Acknowledging the diverse impacts of problem-solving skills on distinct facets of suicide (ideation, attempts, death), health policies should tailor interventions, accounting for individual risk factors and needs to optimize efficacy. Equipping healthcare professionals—therapists, counselors, educators—with problem-solving strategies can bolster their ability to aid at-risk individuals and foster improved mental health outcomes. These recommendations should be contextualized within existing mental health policies, available resources, and specific population requirements, with ongoing research to gauge effectiveness and adaptability as essential components.

## Conclusion

This meta-analysis suggests that high problem-solving skills are associated with a lower probability of suicidal ideation, suicide attempts, and suicide death. Improving problem-solving skills may have a protective effect against suicidal behaviors. However, there is high between-study heterogeneity for some of these results, and evidence of publication bias for some of the outcomes. Therefore, further research is needed to confirm these findings and to identify the specific factors that contribute to the observed associations. It is important to consider the limitations and potential biases of the study when interpreting these results.

## Supporting information

**S1 Checklist. PRISMA 2020 checklist.**
(DOCX)

**S1 Table. Search strategies.**
(DOCX)

**S2 Table. The detailed risk of bias assessment.**
(DOCX)

**S1 Dataset.**
(RM5)

## Acknowledgments

These results were obtained as part of a PhD thesis in Phycology. We would like to appreciate the Vice-Chancellor for Research and Technology of Azad University for approval of this study.

## Author Contributions

**Conceptualization:** Nahid Darvishi, Mehran Farhadi, Jalal Poorolajal.

**Data curation:** Nahid Darvishi, Bita Azmi-Naei, Jalal Poorolajal.

**Formal analysis:** Nahid Darvishi, Jalal Poorolajal.

**Investigation:** Jalal Poorolajal.

**Methodology:** Nahid Darvishi, Mehran Farhadi, Jalal Poorolajal.

**Software:** Jalal Poorolajal.

**Supervision:** Mehran Farhadi, Jalal Poorolajal.

**Validation:** Nahid Darvishi, Bita Azmi-Naei, Jalal Poorolajal.

**Writing – original draft:** Jalal Poorolajal.

**Writing – review & editing:** Nahid Darvishi, Mehran Farhadi, Bita Azmi-Naei, Jalal Poorolajal.

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
