## [Decision Letter · Decision Letter 0]

26 Jul 2023

PONE-D-23-15949The role of problem-solving skills in the prevention of suicidal behaviors: A systematic review and meta-analysisPLOS ONE

Dear Dr. Poorolajal,

Thank you for submitting your manuscript to PLOS ONE. After careful consideration, we feel that it has merit but does not fully meet PLOS ONE’s publication criteria as it currently stands. Therefore, we invite you to submit a revised version of the manuscript that addresses the points raised during the review process.

We look forward to receiving your revised manuscript.

Kind regards,

Humayun Kabir

Academic Editor

PLOS ONE

Journal Requirements:

2. Thank you for stating the following financial disclosure: "X"

3. Thank you for stating the following in your Competing Interests section: "X" 

Reviewers' comments:

Reviewer's Responses to Questions

**Comments to the Author**

1. Is the manuscript technically sound, and do the data support the conclusions?

Reviewer #1: Partly

Reviewer #2: Partly

2. Has the statistical analysis been performed appropriately and rigorously? 

Reviewer #1: Yes

Reviewer #2: No

3. Have the authors made all data underlying the findings in their manuscript fully available?

Reviewer #1: Yes

Reviewer #2: Yes

4. Is the manuscript presented in an intelligible fashion and written in standard English?

Reviewer #1: Yes

Reviewer #2: Yes

5. Review Comments to the Author

Reviewer #1: This paper makes a welcome contribution to understanding about problem solving skills and suicidal behaviour. At present, however, it is somewhat lacking in detail, and I strongly encourage the author/s to consider the suggestions below.

First and foremost, in a full length research article I would expect to see more depth and sophistication in the introduction. There is an opportunity to discuss ways in which different types of problem solving ability are thought to relate to different suicidal behaviors (and why), to elaborate on the different concepts raised and/or to give examples of some past studies (and how conflicts in previous results point to the need for meta-analysis), and to touch on some of the reasons for why the current study used a fairly broad focus (i.e., the paper can be framed as an initial step towards finding points of consensus and difference between existing studies, while also acknowledging that the variation in existing work in terms of definitions, measurement etc makes it challenging to directly compare those studies).

I would also like to see more information about the selection criteria and study selection procedures. For example, what was considered a study on problem solving, vs what was not considered problem solving?

I am also concerned that at a few points, causality seems to be inferred. While it is certainly fair to say that the evidence is very suggestive of a protective effect of problem solving skills, it is not clear at this stage how and why that may be operating (for example, potential mediators or moderators, confounds, etc). This is especially relevant in light of the diverse nature of the different studies included in the meta-analysis.

Finally, I would very much like to see some discussion about the differences between the studies that were included and how and why that may relate to the results.

Reviewer #2: 1. Since WHO reports in 2019 (ref 2) has been listed and cited, it is redundant to list and cite 2014 report (ref 1). Relevant data has been updated in the 2019 report.

2. My major concern is that the conclusion is not sufficiently supported by findings. Findings of RCT studies argued that the problem sovling skills could NOT reduce suicide risk (small effective size), although results of observational studies substantially supported such association. Might such association be attributed to, at least partly, other factors which did not assessed in this study.

3. In the table 1, 17 studies focused on suicidal ideation, 9 on suicide attempt, why in the Figure 2, only 6 and 7 studies have been included? What about the other 13 studies?

4.Only one study focused on suicide death, it is meaningless to review or meta-analyze one study.

5.Authors did not describe any data or information on how to assess so-called problem solving skill, and measure tool or methods were not included in assessing the source of heterogeneity, why? Good psychometric performance and consistent measures are important to assess research findings. Different tools or cut-off points might result in different findings.

6. PLOS authors have the option to publish the peer review history of their article (what does this mean?). If published, this will include your full peer review and any attached files.

Reviewer #1: No

Reviewer #2: No

<quillbot-extension-portal></quillbot-extension-portal>

---

## [Author Response · Author response to Decision Letter 0]

29 Jul 2023

Reviewers' comments

Reviewer #1

This paper makes a welcome contribution to understanding about problem solving skills and suicidal behaviour. At present, however, it is somewhat lacking in detail, and I strongly encourage the author/s to consider the suggestions below.

First and foremost, in a full length research article I would expect to see more depth and sophistication in the introduction. There is an opportunity to discuss ways in which different types of problem solving ability are thought to relate to different suicidal behaviors (and why), to elaborate on the different concepts raised and/or to give examples of some past studies (and how conflicts in previous results point to the need for meta-analysis), and to touch on some of the reasons for why the current study used a fairly broad focus (i.e., the paper can be framed as an initial step towards finding points of consensus and difference between existing studies, while also acknowledging that the variation in existing work in terms of definitions, measurement etc makes it challenging to directly compare those studies).

Answer: Thank you for your feedback. We agree that a full-length research article should have a more comprehensive and sophisticated introduction, and we appreciate the opportunity to revise ours. We carefully revised the introduction section to provide a thorough review of the literature, outline the rationale for the study, and provide context for the research question. We also discussed the existing evidence and gaps in the literature related to the relationship between problem-solving ability and suicidal behaviors. We hope that our revised introduction provides a clearer and more comprehensive understanding of the importance and relevance of our study.

I would also like to see more information about the selection criteria and study selection procedures. For example, what was considered a study on problem solving, vs what was not considered problem solving?

Answer: In the methods section, we provided a detailed explanation of the selection criteria used and presented a definition of problem-solving skills along with the measurement tools that were utilized.

I am also concerned that at a few points, causality seems to be inferred. While it is certainly fair to say that the evidence is very suggestive of a protective effect of problem solving skills, it is not clear at this stage how and why that may be operating (for example, potential mediators or moderators, confounds, etc). This is especially relevant in light of the diverse nature of the different studies included in the meta-analysis.

Answer: We revised the conclusion sections in both the abstract and main text, and modified the final inference based on our findings. Also, in the discussion section, we highlighted the limitations and potential biases of the study.

Finally, I would very much like to see some discussion about the differences between the studies that were included and how and why that may relate to the results.

Answer: In the third paragraph of the discussion section, we provided a detailed explanation of the differences between the studies included in the meta-analysis and how these differences may have impacted the overall results.

Reviewer #2

1. Since WHO reports in 2019 (ref 2) has been listed and cited, it is redundant to list and cite 2014 report (ref 1). Relevant data has been updated in the 2019 report.

Answer: We have replaced the old reference with a new one and revised the introduction accordingly.

2. My major concern is that the conclusion is not sufficiently supported by findings. Findings of RCT studies argued that the problem sovling skills could NOT reduce suicide risk (small effective size), although results of observational studies substantially supported such association. Might such association be attributed to, at least partly, other factors which did not assessed in this study.

Answer: We revised the conclusion sections in both the abstract and main text.

3. In the table 1, 17 studies focused on suicidal ideation, 9 on suicide attempt, why in the Figure 2, only 6 and 7 studies have been included? What about the other 13 studies?

Answer: Thank you for your attention. 

Out of the 27 studies included, 17 focused on suicidal ideation, 9 on suicide attempt, and 1 on suicide death.

Within the 17 studies related to suicidal ideation, 5 reported OR (upper part of Figure 2), 10 reported mean difference (shown in Figure 3), and 2 were RCTs (shown in Figures 2 & 4). Therefore, the total number of studies related to suicidal ideation was 17 (5+10+2).

Within the 9 studies related to suicide attempt, 6 reported OR (shown in Figure 2) and 3 were RCTs (shown in Figures 2 & 4). Therefore, the total number of studies related to suicide attempt was 9 (6+3).

4. Only one study focused on suicide death, it is meaningless to review or meta-analyze one study.

Answer: We appreciate the reviewer's concern about the inclusion of only one study that focused on suicide death. While we agree that it is not possible to conduct a meaningful meta-analysis with only one study, we decided to include this study in our systematic review to provide a comprehensive assessment of the existing literature on the relationship between social support and suicidal behaviors. The study in question is the only case-control study that provides valuable information on the association between social support and suicide death. We acknowledge that further research is needed to confirm these findings, and we have emphasized in the conclusion section the need for additional studies to explore the complex relationship between social support and suicidal behaviors.

5. Authors did not describe any data or information on how to assess so-called problem solving skill, and measure tool or methods were not included in assessing the source of heterogeneity, why? Good psychometric performance and consistent measures are important to assess research findings. Different tools or cut-off points might result in different findings.

Answer: In the method section, we provided an explanation of problem-solving skills and presented a comprehensive list of tools used to measure them in Table 1. As part of our meta-regression analysis, we also included measurement tools to investigate potential sources of heterogeneity.

---

## [Decision Letter · Decision Letter 1]

15 Aug 2023

PONE-D-23-15949R1The role of problem-solving skills in the prevention of suicidal behaviors: A systematic review and meta-analysisPLOS ONE

Dear Dr. Poorolajal,

Thank you for submitting your manuscript to PLOS ONE. After careful consideration, we feel that it has merit but does not fully meet PLOS ONE’s publication criteria as it currently stands. Therefore, we invite you to submit a revised version of the manuscript that addresses the points raised during the review process.

We look forward to receiving your revised manuscript.

Kind regards,

Humayun Kabir

Academic Editor

PLOS ONE

Additional Editor Comments:

Please revise the interpretation of the pooled estimate as the statistical significance should be not emphasized in the meta-analysis.

limitation section and a recommendation section are missing.

is it only search strategy, "The following

keywords were searched: (problem solving OR problem-solving) AND (suicide OR suicidal OR

suicidality)"? If yes, then the search is very narrow and may be no longer systematic research. Describe the details of the search like if you used the MES heading and keywords both. The search strategies should be provided in a supplementary file.

EMBASE and other databases were not covered which may be the potential limitation of the review.

how was the RCT incorporated in this study along with other designs?

"randomized clinical trials (RCT) addressing

the association between problem-solving skills and suicide behaviors" RCT does not incorporate association.

use subheadings while reporting the results.

"problem-solving skills" how this intervention was measured throughout the study?

inclusion and exclusion are missing.

"Control: In cases where studies categorized problem-solving skills, we examined the possibility of

suicidal behaviors in individuals with high problem-solving skills compared to those with low problem solving skills as a control group." what does it mean? specify who is the comparator. if the intervention is problem-solving skills then the comparator should be something else, maybe not having the skill.

"In cases where studies reported the mean score of problem-solving

skills, we examined the standard mean difference between those with and without suicidal behaviors." SMD of which groups? does it a suicidal idea vs no suicidal ideation? if yes, then I think, it is a serious gap in the planning.

cross-sectional study and RCT can not be pooled in same study and should have followed different tools for doing the risk of bias and certainty of the evidence. meta-analysis should not be pooling apple and orange.

"This meta-analysis was conducted to determine to

what extent problem-solving skills can prevent suicidal behaviors" if this is the objective of this review and then I think this review failed to achieve the objective.

Reviewers' comments:

Reviewer's Responses to Questions

**Comments to the Author**

1. If the authors have adequately addressed your comments raised in a previous round of review and you feel that this manuscript is now acceptable for publication, you may indicate that here to bypass the “Comments to the Author” section, enter your conflict of interest statement in the “Confidential to Editor” section, and submit your "Accept" recommendation.

Reviewer #1: All comments have been addressed

2. Is the manuscript technically sound, and do the data support the conclusions?

Reviewer #1: Yes

3. Has the statistical analysis been performed appropriately and rigorously? 

Reviewer #1: Yes

4. Have the authors made all data underlying the findings in their manuscript fully available?

Reviewer #1: Yes

5. Is the manuscript presented in an intelligible fashion and written in standard English?

Reviewer #1: Yes

6. Review Comments to the Author

Reviewer #1: (No Response)

7. PLOS authors have the option to publish the peer review history of their article (what does this mean?). If published, this will include your full peer review and any attached files.

Reviewer #1: No

---

## [Author Response · Author response to Decision Letter 1]

21 Aug 2023

Additional Editor Comments

Thank you for submitting your manuscript to PLOS ONE. After careful consideration, we feel that it has merit but does not fully meet PLOS ONE’s publication criteria as it currently stands. Therefore, we invite you to submit a revised version of the manuscript that addresses the points raised during the review process.

Please revise the interpretation of the pooled estimate as the statistical significance should be not emphasized in the meta-analysis.

Answer: We omitted the terms "significant" and "significantly" from both the abstract and the main results and discussion sections, where we highlighted the outcomes of the meta-analysis.

Limitation section and a recommendation section are missing.

Answer: These sections were added to the end of the Discussion section.

Is it only search strategy, "The following keywords were searched: (problem solving OR problem-solving) AND (suicide OR suicidal OR suicidality)"? If yes, then the search is very narrow and may be no longer systematic research. Describe the details of the search like if you used the MES heading and keywords both. The search strategies should be provided in a supplementary file.

Answer: We have diligently expanded upon our search strategy and incorporated the latest search findings up until August 16, 2023. Our search encompassed a range of pertinent keywords, including terms such as (problem solving or coping skills or resilience skills or solution-oriented skills or cognitive flexibility or decision-making abilities or critical thinking or creative thinking or analytical thinking) and (suicide or suicidal or suicidality). It is noteworthy that both MeSH terms and keywords were thoughtfully integrated into our comprehensive search approach. Fortuitously, the implementation of the new search strategy yielded two additional studies, culminating in a total of 29 studies now included in our analysis. The detailed outline of our search strategies can be found in the supplementary file, providing a clear reference for the methods employed.

EMBASE and other databases were not covered which may be the potential limitation of the review.

Answer: Regrettably, we did not have access to the EMBASE database during our research phase. Nonetheless, a silver lining exists as both the EMBASE and Scopus databases fall under the umbrella of Elsevier Publications. This interconnection ensures that a substantial portion of the resources cataloged in EMBASE are encompassed within the scope of Scopus. Notably, Mann et al. conducted a thorough investigation in 2016 [1] aimed at evaluating coverage, overlap, and distinctive contributions of EMBASE and Scopus. Their findings revealed that 32 of the 34 studies included were sourced from either the EMBASE or Scopus searches. These results offer a compelling insight into the interplay between these databases.

[1] Mann M, Hood K, Trubey R, Powell C. A tale of two databases: a comparison of Embase versus Scopus. In: Challenges to evidence-based health care and Cochrane. Abstracts of the 24th Cochrane Colloquium; 2016 23-27 Oct; Seoul, Korea. John Wiley & Sons; 2016.

How was the RCT incorporated in this study along with other designs? Randomized clinical trials (RCT) addressing the association between problem-solving skills and suicide behaviors" RCT does not incorporate association. Use subheadings while reporting the results.

Answer: We acknowledge the distinct nature of observational studies and randomized controlled trials (RCTs), and we concur that they should be treated as separate entities when consolidating data. To address this valid concern, we have been diligent in our approach, ensuring that each study type receives the due attention it warrants. In the "Methodological Quality" subsection within the Methods section of our manuscript, we have clearly outlined our strategy. Specifically, we employed the Newcastle-Ottawa Scale (NOS) to assess the quality of observational studies and utilized the Delphi List to evaluate the quality of RCTs. Furthermore, we have taken into account the differentiation between the outcomes of observational studies and RCTs in our results. As a result, we have represented the results of observational studies in Figures 2 & 4 and RCTs in Figures 3 & 5. We deeply value your insights and have diligently addressed the issues raised to ensure the accuracy and clarity of our manuscript.

"Problem-solving skills" how this intervention was measured throughout the study?

Answer: In our study, the assessment of “problem-solving skills” was carried out through the utilization of dedicated questionnaires that have been developed and validated for the purpose of evaluating problem-solving abilities. These questionnaires have been widely recognized and accepted within the field as reliable tools for assessing the proficiency of individuals in problem-solving. We addressed this matter by incorporating an explanatory note at the outset of the Methods section to provide clarification. Additionally, a comprehensive listing of these questionnaires is available in Table 1 for reference.

Inclusion and exclusion are missing.

Answer: The inclusion and exclusion criteria for our meta-analysis were defined using the PICOS framework, which encompasses Population, Intervention/Exposure, Comparison, Outcome, and Study design characteristics. These criteria were systematically applied to identify relevant studies for our analysis. We incorporated these criteria at the outset of the Methods section to provide a clear understanding of our study's criteria for study selection.

“Control: In cases where studies categorized problem-solving skills, we examined the possibility of suicidal behaviors in individuals with high problem-solving skills compared to those with low problem solving skills as a control group.” What does it mean? Specify who is the comparator. If the intervention is problem-solving skills then the comparator should be something else, maybe not having the skill.

Answer: Certainly, I appreciate your feedback. In our study, we adopted two distinct approaches to investigate the association between problem-solving skills and suicidal behaviors.

Firstly, in cases where studies categorized individuals based on their problem-solving skills (high vs. low), we examined the odds of suicidal behaviors in those with high problem-solving skills compared to those with low problem-solving skills (reference group). This allowed us to assess the impact of varying problem-solving skill levels on suicidal behaviors.

Secondly, we encountered studies that measured the mean problem-solving skill scores among individuals with and without suicidal behaviors. In these cases, we calculated the standardized mean difference (SMD) of the problem-solving skill scores between these two groups. This approach enabled us to explore the magnitude of difference in problem-solving abilities between individuals exhibiting suicidal behaviors and those who did not.

To present these distinct analyses accurately, we reported the pooled odds ratios for the first approach and the SMD values for the second approach separately. By employing these two different methodologies, we aimed to comprehensively capture the relationship between problem-solving skills and suicidal behaviors, providing a more nuanced understanding of their interplay. We clarified these points further in revised manuscript for better clarity and transparency.

“In cases where studies reported the mean score of problem-solving skills, we examined the standard mean difference between those with and without suicidal behaviors.” SMD of which groups? Does it a suicidal idea vs no suicidal ideation? If yes, then I think, it is a serious gap in the planning.

Answer: Thank you for your feedback. We have taken your current comments into consideration and addressed the concerns raised in response to your previous comment. We are committed to ensuring the clarity and accuracy of our manuscript, and we have incorporated the suggestions and considerations discussed earlier. We appreciate your guidance and are dedicated to producing a thorough and well-structured manuscript that adheres to the highest standards of research integrity.

Cross-sectional study and RCT cannot be pooled in same study and should have followed different tools for doing the risk of bias and certainty of the evidence. Meta-analysis should not be pooling apple and orange.

Answer: We appreciate the editor's observation. We completely understand the importance of maintaining methodological rigor when conducting a meta-analysis. We agree that observational studies and randomized controlled trials (RCTs) are distinct in nature and should be treated as separate entities when pooling data. To address this concern, we would like to highlight that we were meticulous in our approach to ensure that each study type received appropriate attention. Specifically, under the "Methodological Quality" subsection in the Methods section of our manuscript, we clearly delineated our approach. For observational studies, we employed the Newcastle-Ottawa Scale (NOS) to assess their quality, while for RCTs, we used the Delphi List to evaluate their quality. This approach enabled us to rigorously evaluate the risk of bias and the certainty of evidence for each study type, ensuring that the distinct characteristics of observational studies and RCTs were appropriately considered. Certainly, we recognize the importance of maintaining a clear distinction between the results of observational studies and RCTs within our figures. To address this, the results of observational studies are depicted separately in Figures 2 & 4 and that of and RCTs in Figures 3 and 5. 

“This meta-analysis was conducted to determine to what extent problem-solving skills can prevent suicidal behaviors”. If this is the objective of this review and then I think this review failed to achieve the objective.

Answer: Thank you for your feedback. We appreciate your insights, and we have carefully considered your comments. In response to your concern, we have revised the objective of the meta-analysis to ensure clarity and alignment with the study's findings. The modified objective now reads: “This meta-analysis was conducted to assess the association between problem-solving skills and suicidal behaviors and elucidate the potential role of problem-solving skills in influencing the occurrence of suicidal behaviors.”

We believe that this modified objective better captures the focus and outcomes of our study, and we are grateful for your input in refining our work.

---

## [Decision Letter · Decision Letter 2]

18 Sep 2023

PONE-D-23-15949R2The role of problem-solving skills in the prevention of suicidal behaviors: A systematic review and meta-analysisPLOS ONE

Dear Dr. Poorolajal,

Thank you for submitting your manuscript to PLOS ONE. After careful consideration, we feel that it has merit but does not fully meet PLOS ONE’s publication criteria as it currently stands. Therefore, we invite you to submit a revised version of the manuscript that addresses the points raised during the review process.

We look forward to receiving your revised manuscript.

Kind regards,

Humayun Kabir

Academic Editor

PLOS ONE

Journal Requirements:

Reviewers' comments:

Reviewer's Responses to Questions

**Comments to the Author**

1. If the authors have adequately addressed your comments raised in a previous round of review and you feel that this manuscript is now acceptable for publication, you may indicate that here to bypass the “Comments to the Author” section, enter your conflict of interest statement in the “Confidential to Editor” section, and submit your "Accept" recommendation.

Reviewer #1: All comments have been addressed

Reviewer #3: (No Response)

2. Is the manuscript technically sound, and do the data support the conclusions?

Reviewer #1: Yes

Reviewer #3: Yes

3. Has the statistical analysis been performed appropriately and rigorously? 

Reviewer #1: Yes

Reviewer #3: Yes

4. Have the authors made all data underlying the findings in their manuscript fully available?

Reviewer #1: Yes

Reviewer #3: Yes

5. Is the manuscript presented in an intelligible fashion and written in standard English?

Reviewer #1: Yes

Reviewer #3: Yes

6. Review Comments to the Author

Reviewer #1: (No Response)

Reviewer #3: Thanks for giving me the opportunity to review this manuscript. After careful revision of this manuscript, I suggest the following minor edits -

1. Methods: the heterogeneity and publication bias section should be reported below the Data synthesis section.

2. Authors are suggested to report the detailed risk of bias assessment in the supplementary file.

3. The recommendation section should be merged with the discussion section without mentioning the heading "Recommendation"

7. PLOS authors have the option to publish the peer review history of their article (what does this mean?). If published, this will include your full peer review and any attached files.

Reviewer #1: No

Reviewer #3: **Yes: **Saifur Rahman Chowdhury

---

## [Author Response · Author response to Decision Letter 2]

19 Sep 2023

Reviewer #3

Thanks for giving me the opportunity to review this manuscript. After careful revision of this manuscript, I suggest the following minor edits.

1. Methods: the heterogeneity and publication bias section should be reported below the Data synthesis section.

Response: Thank you for your valuable feedback and for your careful review of our manuscript. We have carefully considered your suggestion regarding the placement of the "Heterogeneity and publication bias" section in our manuscript. Following your recommendation, we have moved this section below the "Data synthesis" section to enhance the logical flow and organization of our paper.

2. Authors are suggested to report the detailed risk of bias assessment in the supplementary file.

Response: Regarding your suggestion to report the detailed risk of bias assessment in the supplementary file, we have taken this recommendation into account. We have included a comprehensive risk of bias assessment in the supplementary material, providing a detailed and transparent account of our evaluation process.

3. The recommendation section should be merged with the discussion section without mentioning the heading "Recommendation"

Response: In accordance with your recommendation, we have removed the heading "Recommendation" and integrated the content seamlessly into the broader "Discussion" section.

---

## [Decision Letter · Decision Letter 3]

17 Oct 2023

The role of problem-solving skills in the prevention of suicidal behaviors: A systematic review and meta-analysis

PONE-D-23-15949R3

Dear Dr. Poorolajal,

We’re pleased to inform you that your manuscript has been judged scientifically suitable for publication and will be formally accepted for publication once it meets all outstanding technical requirements.

Kind regards,

Humayun Kabir

Academic Editor

PLOS ONE

Additional Editor Comments (optional):

Reviewers' comments:

Reviewer's Responses to Questions

**Comments to the Author**

1. If the authors have adequately addressed your comments raised in a previous round of review and you feel that this manuscript is now acceptable for publication, you may indicate that here to bypass the “Comments to the Author” section, enter your conflict of interest statement in the “Confidential to Editor” section, and submit your "Accept" recommendation.

Reviewer #1: All comments have been addressed

Reviewer #3: All comments have been addressed

2. Is the manuscript technically sound, and do the data support the conclusions?

Reviewer #1: (No Response)

Reviewer #3: Yes

3. Has the statistical analysis been performed appropriately and rigorously? 

Reviewer #1: (No Response)

Reviewer #3: Yes

4. Have the authors made all data underlying the findings in their manuscript fully available?

Reviewer #1: (No Response)

Reviewer #3: (No Response)

5. Is the manuscript presented in an intelligible fashion and written in standard English?

Reviewer #1: (No Response)

Reviewer #3: Yes

6. Review Comments to the Author

Reviewer #1: (No Response)

Reviewer #3: (No Response)

7. PLOS authors have the option to publish the peer review history of their article (what does this mean?). If published, this will include your full peer review and any attached files.

Reviewer #1: No

Reviewer #3: No

---

## [Editor Report · Acceptance letter]

23 Oct 2023

PONE-D-23-15949R3 

The role of problem-solving skills in the prevention of suicidal behaviors: A systematic review and meta-analysis 

Dear Dr. Poorolajal:

I'm pleased to inform you that your manuscript has been deemed suitable for publication in PLOS ONE. Congratulations! Your manuscript is now with our production department. 

Kind regards, 

on behalf of

Dr. Humayun Kabir 

Academic Editor

PLOS ONE